# Enhancing Corrosion and Wear Resistance of Ti6Al4V Alloy Using CNTs Mixed Electro-Discharge Process

**DOI:** 10.3390/mi11090850

**Published:** 2020-09-12

**Authors:** Gurpreet Singh, Timur Rizovich Ablyaz, Evgeny Sergeevich Shlykov, Karim Ravilevich Muratov, Amandeep Singh Bhui, Sarabjeet Singh Sidhu

**Affiliations:** 1Mechanical Engineering Department, Beant College of Engineering and Technology, Gurdaspur 143521, India; singh.gurpreet191@gmail.com (G.S.); meet_amandeep@yahoo.com (A.S.B.); sarabjeetsidhu@yahoo.com (S.S.S.); 2Mechanical Engineering Faculty, Perm National Research Polytechnic University, 614000 Perm, Russia; kruspert@mail.ru (E.S.S.); karimur_80@mail.ru (K.R.M.)

**Keywords:** electro-discharge treatment, Ti-6Al-4V, MWCNTs, surface characterization, wear resistance, corrosion resistance

## Abstract

This paper presents wear and corrosion resistance analysis of carbon nanotubes coated with Ti-6Al-4V alloy processed by electro-discharge treatment. The reported work is carried out using Taguchi’s L18 orthogonal array to design the experimental matrix by varying five input process parameters i.e., dielectric medium (plain dielectric, multi-walled carbon nanotubes (MWCNTs) mixed dielectric), current (1–4 A), pulse-on-time (30–60 µs), pulse-off-time (60–120 µs), and voltage (30–50 V). The output responses are assessed in terms of microhardness and surface roughness of the treated specimen. X-ray diffraction (XRD) spectra of the coated sample reveal the formation of intermetallic compounds, oxides, and carbides, whereas surface morphology is observed using scanning electron microscopy (SEM) analysis. For the purpose of the in-vitro wear behavior of treated samples, the surface with superior microhardness values in plain dielectric and MWCNTs mixed dielectric is compared using a pin-on-disc type wear test. Furthermore, electrochemical corrosion test is also conducted to portray the dominance of treated substrate of Ti-6Al-4V alloy for biomedical applications. It is concluded that the wear-resistant and the corrosion protection efficiency of the MWCNTs treated substrate enhanced to 95%, and 96.63%, respectively.

## 1. Introduction

Progress into medical diagnosis, therapy, and rehabilitation is not achievable without the persistent advances in the development of novel or refined materials. Evolutions in the field of metallic biomaterials have remarkably contributed to orthopedic, dental, cardiovascular, neural, and urological praxis [1,2]. Ti-6Al-4V alloy is very favorable in orthopedic applications owing to high weight-to-strength ratio, corrosion resistance, and biocompatibility. Yet, it possesses inferior wear and abrasion resilience, due to low hardness, and releases of toxic ions from the alloy surface within the human body environment [3]. The rather sub-standard tribo-characteristics and corrosion performance, possessed by Ti-6Al-4V alloy, has prompted the progress of various surface treatment approaches that include physical vapor deposition, chemical vapor deposition, dip coating, ion implantation, sol-gel, thermal treatments, laser alloying, etc. [4,5].

Among all the frequently-used techniques for surface-treatment, electro-discharge treatment (EDT) is a thermoelectric method that unfolded as a potential method for the surface modification process [6,7]. As a progressive technique in modifying substrates, EDT widely used to improve the surface characteristics, chemical, mechanical, corrosion, and tribological behavior of the base material [8]. In this process, the train of sparks within the electrodes resulted in the modification of substrate surface due to the material transfer mechanism [9,10,11]. During the EDT process, the particles of desired coating powder, added in the dielectric medium suspended around the discharge column, accelerated and gained sufficient velocity to penetrate to the molten pool before solidification by means of electrophoresis and negative pressure. This was induced after cessation of a discharge, which leads a surface embedded with added fine particles [12]. This technique significantly affects the surface characteristics of the modified surface, such as morphological structure and chemical compositional analysis [13]. Moreover, the surface produced by EDT exhibited superior corrosion-resistance, wear-resistance as compared to the untreated substrate material, and promoted bioactivity [14].

Various powders have been used by the researchers to investigate their machining performance and influence on the surface characteristics of the machined surface. The most commonly utilized powders are silicon, aluminum, chromium, silicon carbide, titanium, tantalum, boron carbide, and carbon nanotubes [15]. The selection of powder in the dielectric medium significantly depends on the final application of the product. For instance, carbon nanotubes exist in two variants, i.e., single-walled carbon nanotubes (SWCNTs) and multi-walled carbon nanotubes (MWCNTs). The MWCNTs are concentrically aligned sheets of graphene in the shape of cylinders having an outer diameter in the range of 20 to 50 nm [16]. These MWCNTs have been in the limelight in recent decades for exhibiting superior properties pertaining to chemical, electrical, mechanical durability, strength, and bioactivity [17,18]. As a result, carbon nanotubes prominently found applications in the area of medicine and biomedical industries [19,20].

### Related Work

This section presents the overview of the research work associated with the surface modification of Ti-6Al-4V alloy for improved wear-resistance, microhardness, corrosion-resistance, and bioactivity. Sarraf et al. [21] reported the improved biological responses of Ti-6Al-4V alloy using the physical vapor deposition method. They had used the tantalum pentoxide nanotube for the surface modification of base metal and observed remarkable nanomechanical properties, wear-resistant, and corrosion-resistant surface with a protective efficiency of 95% in comparison to the base metal. In another study, plasma surface alloying was employed by Li et al. [22] to enhance the corrosion, and wear resistance of Ti-6Al-4V by forming a coated layer of ZrO_2_/TiO_2_ for biomedical applications. They validated the significance of the modified surface on their excellent performance in wear and corrosion testing. Pogrebnjak et al. [23] demonstrated that the implantation of W and Mo ions, followed by annealing of Ti-6Al-4V surface, resulted in improved nano-hardness and wear resilience, due to the formation of nitrides, carbo-nitrides, and inter-metalloids. Man et. al. [24] enhanced the tribological properties of Ti-6Al-4V alloy using laser diffusion nitriding. They found that the micro-hardness was increased by 2.3 times and wear resilience increased by 8 times compared to the substrate. Kgoete et al. [25] used the spark plasma sintering method to process the Ti-6Al-4V alloy reinforced with µ-sized Si_3_N_4_ powder to evaluate its corrosion behavior using electrochemical corrosion analysis. Numerous bioactive powders were used by the researchers to enhance the surface competency of Ti-6Al-4V alloy for biomedical applications [26]. However, in a review by Li et al. [27], it was concluded that the coating of CNTs on biomaterials works as a promising combination for achieving satisfactory tissue engineering, hence promoting cell attachment and proliferation. CNTs coating of Ti-6Al-4V alloy demonstrated enhanced fracture toughness, wear-resistance, and biocompatibility making this more favorable for load-bearing orthopedic implants. Deng et al. [28] concluded that the Ti-6Al-4V alloy surface treated with MWCNTs showed superior bio-tribological properties, which is the necessary condition for the bioimplant experiencing various loading conditions within the body. Similarly, Terada et al. [29] reported that the titanium alloy, coated with MWCNTs, improved surface roughness, which accelerated cell adhesion and proliferation.

The present article addresses the influence of electro-discharge process on the microhardness, surface roughness, surface morphology, phase transformation, wear, and corrosion behavior of Ti-6Al-4V surface. In the first step, the significant impact of the selected process parameters on the microhardness and surface roughness was analyzed. In the next step, the characterization of EDT modified surface in terms of morphological and compositional analysis was completed by using SEM, and XRD techniques. The Ti-6Al-4V alloy is well-established and a commonly used biomaterial for orthopedics and dental implants, where wear-resistance and corrosion-resistance play a vital role in long-term functioning of the implants. Therefore, pin-on-disc wear test and electrochemical corrosion behavior of treated samples were performed.

## 2. Material and Methods

### 2.1. Material

Commercial grade-5, Ti-6Al-4V (α + β) titanium alloy was purchased in the form of a plate with dimensions 160 × 80 × 5 (units: mm) from Baoji Fuyuantong Industry and Trade Co. Ltd., Baoji, China and used as workpiece material. Electrolytic pure graphite procured from Mersen Germany (Courbevoie, France), machined to 9.5 mm diameter was chosen as a tool electrode. Table 1 shows the physical properties of the workpiece and electrode used in the present experimentation. Table 2 listed the functionalized multi-walled carbon nanotubes (MWCNTs) properties, purchased from United Nanotech, Bangalore, India. These MWCNTs were mixed in commercial-grade hydrocarbon oil (EDM oil) to achieve the desired surface modification of Ti-6Al-4V alloy in EDT process.

### 2.2. Experimental Work

In this experiment, Taguchi’s L18 orthogonal array incorporating mixed-level design (2^1^ × 3^4^) was selected. To draw valid conclusions, the machining parameters were current, pulse-on/off duration and voltage with three levels whereas two types of dielectric medium were selected. The different values for the levels were chosen from the pilot trials performed and reported in our earlier study [30]. Table 3 illustrates the machining parameters and the corresponding levels for the design of experiment.

Table 4, illustrates the array of 18 experimental trials undertaken to investigate the potential of selected machining parameters on output responses of EDT process. For experimental trials, the numerical controlled electrical discharge machine with side flushing mechanism was used (make: OSCARMAX, Taichung City, Taiwan; model: S645 CMAX). The trials were conducted in doublet i.e., a total of 36 experiments (18 × 2 runs) were carried out for more precise output responses. The set of first nine experiments were conducted in plain dielectric type, i.e., EDM oil, followed by remaining experiments in MWCNTs mixed dielectric medium. The concentration of MWCNTs (7 g/L) in the dielectric medium was preferred on the basis of previously reported work by the authors, where higher concentration causes unstable machining [30]. Figure 1 illustrates the indigenously developed dielectric tank (12” × 9” × 9”) for the execution of MWCNTs mixed experiments. A circulation pump and stirrer were introduced for efficient flushing and suspension of nano-powder in the spark gap during machining. The polarity (i.e.,workpiece (–), tool (+)), and machining depth of 1 mm were kept constant for all the experimental runs.

### 2.3. Measurement and Calculations of Output Responses

After the EDT, the output responses were assessed in the terms of surface roughness (SR) and microhardness (MH) of the machined substrate. Microhardness tester (model: HM-220, make: Mitutoyo, Neuss, Germany) and surface roughness tester (model: SJ-401, make: Mitutoyo, Germany) was used to measure the microhardness (units: HV) and surface roughness (units: µm) respectively, by taking three readings at distinct points. The microhardness was measured by profiling a pyramidal imprint on the sample surface using a diamond indenter under low-force hardness scale (HV 0.5) with a test force-load of 4.9 N and for a dwell time of 10 s.

For determining the roughness of EDT surface, stylus method was employed to find the roughness value (Ra) with a profile resolution of 12 nm, and range of 80 μm having resolution of 0.001 μm. It was reported in the literature that higher surface roughness participated in cell anchoring, facilities better bone-implant adhesion due to the presence of micro-crack and pores [31,32,33]. Therefore, for further testing, the samples with higher surface roughness were sectioned from the workpiece by using wire-EDM.

### 2.4. Surface Characterization of EDT specimen

The electro-discharge treated surface was thoroughly cleaned in acetone to remove any foreign particles or EDM oil entrapped in pores. The surface morphology of the sample depicting superior output responses was inspected using a scanning electron microscope (SEM; model: JSM-6610LV, make: Jeol Ltd., Tokyo, Japan) at an accelerating voltage of 20 kV. The phase composition of the EDT surface was examined by X-ray diffractometer (XRD; model: PANalytical X’Pert Pro MPD, make: Panalytical, Almelo, The Netherlands) with Cu-Kα X-ray radiations at λ = 1.5406 Å using 40 mA and 45 kV as generator settings.

### 2.5. Examination of Wear and Electrochemical Corrosion Behavior

Furthermore, in-vitro wear and corrosion tests were executed to scrutinize the wear resistance and corrosion resistance offered by the EDT samples. Two samples were selected in both cases, i.e., with maximum output values in MWCNTs mixed dielectric, in the plain dielectric, and it was further compared with the untreated substrate. The experimental trial depicting higher microhardness value in each dielectric medium (i.e., plain, MWCNTs mixed) was examined for the wear test, and similarly the two samples with higher surface roughness value in each dielectric medium was chosen for the electrochemical corrosion test. A pin-on-disk tribometer (Ducom instruments, Bangalore, India) was used to perform the wear test of specimens on EN31 stainless steel disk with diameter 120 mm, thickness 10 mm, and hardness of 63 HRC. The fixed parameters for the wear test were a steady load of 70 N, speed of rotating disc at 100 revolutions/min, and track diameter of 80 mm. The experimentation was conducted at room temperature using the ringer’s solution (Nice Chemicals Pvt. Ltd., Cochin, India; pH value 7.2) as a lubricant to replicate the human body conditions during the investigation. The wear process of the pins was monitored by an incorporated computer system with TR-20LE software, and the combined plot of all samples was plotted using the reading values in OriginPro 8 software.

The potentiodynamic polarization technique was used to inspect the corrosion behavior of samples and the test was carried out on potentiostat/galvanostat electrochemical instrument (Metrohm Autolab, PGSTAT 302, Utrecht, The Netherlands) at ambient temperature. The equipment was an arrangement of three-electrodes, namely silver/silver chloride (Ag/AgCl) as a reference electrode, platinum rod (Pt) as a counter electrode, and workpiece sample (Ti-6Al-4V) as a working electrode. For better results, the surface of the specimen was covered with insulating tape to prevent the initiation of corrosion and exposed the fixed area of 0.32 cm^2^ for testing. Prior to the testing, the samples were immersed in the solution for 24 h to stabilization. Likewise, for wear test analysis, ringer’s solution is used as simulated body fluid (electrolyte in testing) to imitate the human body environment for the bio-implant. The results of the Tafel exploration plots were analyzed using the Nova software incorporated with the experimental arrangement, and the output values for anodic slope (βa), cathodic slope (βc), corrosion current density (*i*_corr_), corrosion potential (E_corr_) were recorded. Accordingly, the corrosion rate (CR) of the samples was then evaluated according to ASTM G102-89 [34], a standard for electrochemical measurements using Faraday’s law (Equation (1)),
(1)$$\mathrm{CR}=K\;\frac{i_{\mathrm{corr}}}{\rho }\;\mathrm{EW}$$
where
CR is in mm/year,*i*_corr_ is in µA/cm^2^;K = 3.27 × 10^−3^, mm g/µA cm year;ρ = density in g/cm^3^;EW = equivalent weight of the material.

## 3. Results and Discussion

### 3.1. Evaluation of the Output Responses

Table 5 demonstrates the measured values with three repetitions, and signal-to-noise (SN) ratios of microhardness and surface roughness associated with each experimental trial. The evaluation of microhardness and surface roughness values were as per the Taguchi’s criterion “larger-is-better” (Equation (2)), where a higher value of SN ratio represents the favorable response value:(2)$$\mathrm{Larger}-\mathrm{is}-\mathrm{better}:\;\eta \;=\;-10\log\;\left[\frac{1}{n}\sum_{i=1}^{n}{\mathrm{yi}}^{-2}\right].$$
where
‘η’ denotes the SN ratio (dB);‘yi’ indicates the value of *i*^th^ trial of experimental trial;‘n’ is the repetition of the experiment.

Additionally, the experimentally calculated values were subsequently analyzed via analysis of variance (ANOVA) to inspect the influential input parameters and their significance, in terms of percentage contribution for microhardness, and surface roughness, respectively.

#### 3.1.1. Analysis of Microhardness

In electro-discharge treatment, the powder particles tend to enhance the microhardness by forming a coating layer of various intermetallic compounds, and carbides on the treated substrate [35,36]. Table 6 demonstrates the ANOVA results of microhardness of the machined surface. As shown in Table 6, dielectric type (MWCNTs mixed) noticed as the most eminent factor with the contribution of 83.019%, followed by current with the contribution of 13.698% affecting the microhardness of machined substrate.

Similar results were revealed by the signal-to-noise ratio plot (Figure 2), disclosing dielectric type, current, pulse-on-time as momentous parameters, and pulse-off-time, voltage as insignificant parameters for the microhardness of treated samples. The higher intensity of the current and pulse-on-time steeply promotes the deposition of particles, which improves the microhardness, and other changes in surface characteristics of the machining area during the EDT process [37,38,39]. Based on the results, higher values of current (4A) and MWCNTs, mixed dielectric medium, depicts utmost mean microhardness (4452.5 HV, trial 17) with an increase of 2.5 times comparative to the sample treated in plain dielectric (1765.2 HV, trial 9), and 10 times superior to the untreated material (435.4 HV), respectively.

#### 3.1.2. Analysis of Surface Roughness

The ANOVA Table 7 manifested current (contribution: 61.076%) and dielectric type (contribution: 16.551%) as the prominent factors affecting the surface roughness of EDT processed Ti-6Al-4V substrate. As cleared from the SN ratios plot, shown in Figure 3, MWCNTs mixed dielectric and 4A of current intensity were the dominant parameters for the roughness of the EDT surface. In contrast, the variation of signal-to-noise ratios for pulse-on/off duration, voltage was not influential and termed as in-significant factors. Moreover, the interaction between dielectric type and current also noticed as significant, having *p*-value 0.001 and a contribution of 19.824%.

Figure 4 portrays the variation of surface roughness in conjunction with the dielectric type and current. It was observed that current at level 3 (4 A) along with MWCNTs mixed dielectric increases the surface roughness of the treated samples. Whereas, at the lower level (1 A) of current intensity, changing the dielectric type from MWCNTs mixed to plain dielectric drastically decreases the surface roughness. Therefore, it can be revealed that the dielectric illustrates a substantial role in the surface roughness of the EDT surface. The maximum mean roughness of 1.240 μm (trial 17) was depicted at parametric settings of 4 A current, 45 μs pulse-on-time, 60 μs pulse-off-time, and 50V of voltage in MWCNTs mixed dielectric medium. Moreover, all the three trials (trial 16–18) in MWCNTs mixed dielectric at a discharge current of 4A exhibits good roughness values within the variation of 4% (approx.). The roughness value reported is acceptable and within the range, required for the bio-implants for proper cell proliferation, bone-implant fixation [40,41].

### 3.2. Morphology and Phase Composition Analysis of MWCNTs Treated Specimen

The supremacy of the carbon nanotubes mixed dielectric was investigated using scanning electron microscopy. Figure 5a,b manifests the morphological of untreated and treated substrate. For morphological examination, sample depicting superior microhardness and higher surface roughness (trial 17) was selected. It is evident from the micrograph (Figure 5b) that a higher spark energy, coupled with carbon particles, produce micro-macropores, small crests on the EDT surface. The existence of such pores, microcracks, and molten metal droplets on the surface facilitated good adhesion within the implant-bone interface, and offered cell growth for proper regeneration [42,43].

Furthermore, the investigation of the electro-discharge-treated sample with maximum microhardness and surface roughness was carried using an X-ray diffractometer within the range of 20°–80°. The XRD spectra in Figure 6 revealed the formation of various intermetallic compounds during the EDT process, which majorly contributes to improving the microhardness of the treated substrate. The newly formed compounds namely calcite (C_1_Ca_1_O_3_), titanium nickel (Ni_1_T_1_) shows the hexagonal structure, molybdenum germanide (Ge_2_Mo_1_), molybdenum nickel (Mo_1_Ni_4_) represents a tetragonal structure, and aluminium nitride (Al_1_N_1_), chromium oxide (Cr_3_O_1_) disclosed as a cubic structure. Alongside, high peaks of titanium carbide (TiC), vanadium silicide (Si_2_V_1_) were witnessed on the surface, and the presence of such hard-bioactive layers exhibits better resistance to corrosion, wear by providing appropriate implant-bone bioactivity within the individual. Moreover, the MWCNTs-EDT surface was further examined for the valuation of in-vitro behavior to affirm the enhanced wear resistance and corrosion resistance by comparing the results with untreated surface and the sample treated in the plain dielectric.

### 3.3. In-Vitro Wear and Corrosion Analysis of EDT Samples

The wear resistance proficiency of the treated Ti-6Al-4V surface was investigated using a pin-on-disc arrangement. Three samples were selected, i.e., untreated (as received), treated in the plain dielectric (trial 9) and in MWCNTs mixed dielectric (trial 17) for evaluation of wear performance. The samples were chosen on the basis of maximum mean microhardness in plain dielectric (1765.2 HV, trial 9) and MWCNTs mixed dielectric (4452.5 HV, trial 17).

Figure 7 unfolded the combined wear plot for three samples, and it was observed that the wear of substrate was nearly at an even rate of 450 μm. The wear of sample, treated in the plain dielectric, started after 150 μm in the beginning, due to the presence of ridges, then the wear rate became constant to 125 μm (approx.) up to 250 s until the machined or recast layer wore off. Afterwards, the wear rate was raised to 440 μm, which was close to the wear rate of the substrate material. The wear of MWCNTs-treated sample started initially at 50 μm, and once the surface became even from all edges, a constant wear rate of 22 μm was exhibited by the MWCNTs-EDT sample up to 900 s. This wear response was due to the modified surface having intermetallic compounds, carbides, and silicide, as discussed in the section of XRD analysis. Later, the wear rate was abruptly raised once the coated layer wore out, demonstrating an improved resistance to wear by 95% compared to substrate metal.

In the electrochemical test analysis, the surface with maximum roughness value in the plain dielectric (trial 7), and MWCNTs mixed dielectric (trial 17), was compared with the untreated substrate to scrutinize the efficacy of the EDT method. The additional electrochemical characteristics attained via potentiodynamic polarization analysis were tabulated in Table 8.

Figure 8 illustrates the combined polarization curves for the substrate and selected EDT samples. The plot showed that the MWCNTs-coated specimen exhibits a higher value of corrosion potential (E_corr_), and thus, possesses a low corrosion rate compared to other samples. In contrast, the untreated substrate sample showed the lowest corrosion potential (E_corr_), corrosion-resistance, and subsequently the higher corrosion current (*i*_corr_), compared to all the EDT samples. The results of in-vitro corrosion behavior of the compared samples demonstrated that the EDT process uplifts the corrosion-resistant competences of the treated surface.

In Table 8, MWCNTs-EDT specimen exhibit superior resistance to corrosion (CR) and a lesser value of corrosion current density (*i*_corr_). These are the two main factors that directly influence the performance of a material under biological conditions. Also, carbon nanotubes-coated specimens depicted a higher value of corrosion potential (E_corr_ = 3.5086 mV) compared to another tested specimen, i.e., −82.492 mV is recorded for treated in plain dielectric and −160.640 mV for untreated substrate of Ti-6Al-4V alloy.

Additionally, the protection efficiency (P_e_) of the EDT samples was calculated using the Equation (3),
(3)$$P_{e}\left(\%\right)=\left[1-\frac{i_{\mathrm{corr}}}{i_{\mathrm{corr}}{}^{\mathrm{substrate}}}\right]\times 100$$
where, *i*_corr_ denotes the corrosion current densities of EDT samples (in MWCNTs and plain dielectric) to the *i*_corr_ of the substrate sample, respectively.

The upmost protective efficiency of 96.63% was observed for MWCNTs-EDT on Ti-6Al-4V alloy. Finally, it was concluded that the MWCNT-modified Ti-6Al-4V alloy surface revealed enhanced bioactivity by minimizing the discharge of ions (Al and V) from the implant material causing toxicity. Therefore, the responses of in-vitro wear and corrosion analysis confirmed that the electro-discharge treated samples in MWCNT-mixed dielectric considerably discloses the surface for improved wear-resistance and corrosion-resistance.

## 4. Conclusions

The Ti-6Al-4V alloy surface was processed by electro-discharge treatment with the purpose of improving its surface hardness, surface characteristics, and to examine the in-vitro corrosion behavior and tribological performance of the modified surface. In the present work, the treated samples were compared with the substrate sample and the following conclusions can be drawn:

The surface characterization of the samples revealed that the surface-treated in MWCNTs medium demonstrated improved surface hardness to 10 times (4452.5 HV), compared with the untreated substrate sample (435.4 HV), thus, increasing wear-resistance to 95%.

The electrochemical corrosion analysis demonstrated that MWCNTs-EDT surface exhibited higher corrosion potential, and this non-reactive surface was attained because of the exceptional thermal properties and chemical stability of MWCNTs. The newly formed carbide- and oxide-rich layers stimulate the corrosion-resistance of the EDT surface.

The XRD pattern of the EDT sample substantiated the formation of intermetallic compounds (titanium nickel, molybdenum nickel, aluminium nitride, vanadium silicide), oxide (chromium oxide), and carbide (titanium carbide) on the treated surface that encourages the bioactivity, wear-resistance and corrosion-resistance of the EDT surface.

SEM analysis disclosed that MWCNTs-EDT surface with the presence of small craters, and evenly distributed micro-pores on the surface, which may facilitates the proper bone-implant adhesion and promotes cell proliferation.

The surface treatment of Ti alloy in MWCNTs mixed dielectric at 4 A current, 45 µs pulse-on-time, 60 µs pulse-off-time, and 50 V delivered the superlative results of tribological performance and corrosion analysis.

Therefore, the electro-discharge treated titanium-based implants can be further inspected for their practice in clinical applications. Moreover, the reported research can be further protracted to investigate the cytocompatibility, hemocompatibility, surface wettability, toxic ions degradation rate, etc., of the EDT surfaces.

## Figures and Tables

**Figure 1 micromachines-11-00850-f001:**
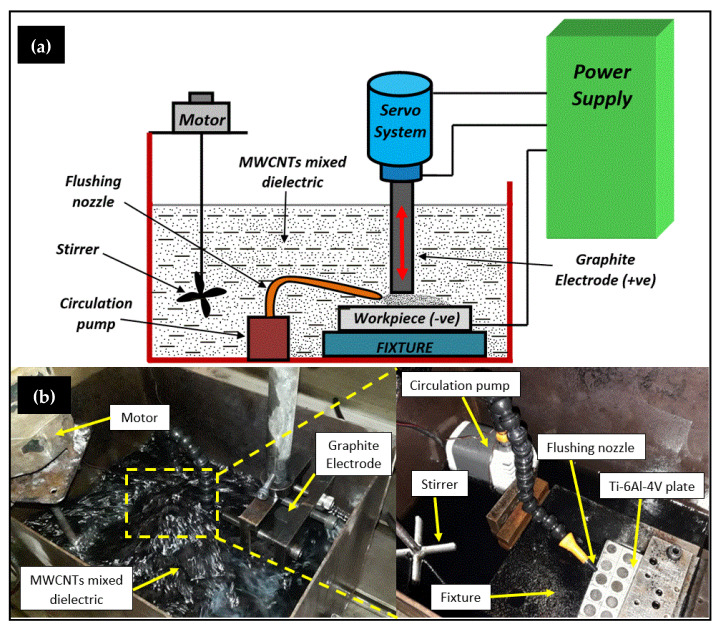
EDT line-diagram in; (**a**), and setup for MWCNTs-EDT in (**b**).

**Figure 2 micromachines-11-00850-f002:**
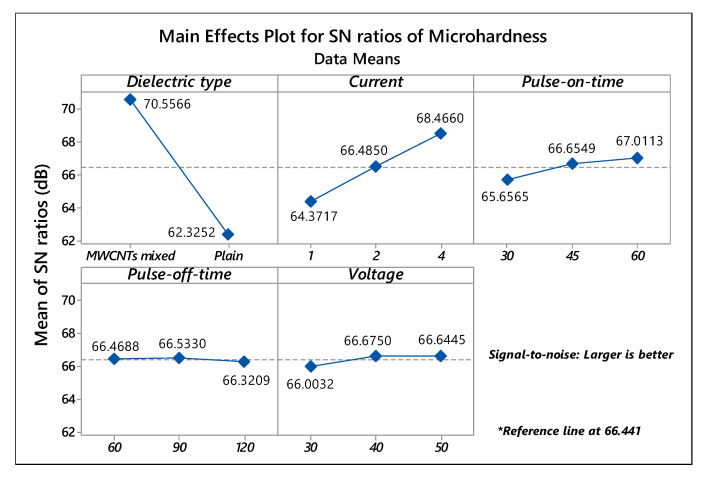
Main effects SN ratios plot of Microhardness.

**Figure 3 micromachines-11-00850-f003:**
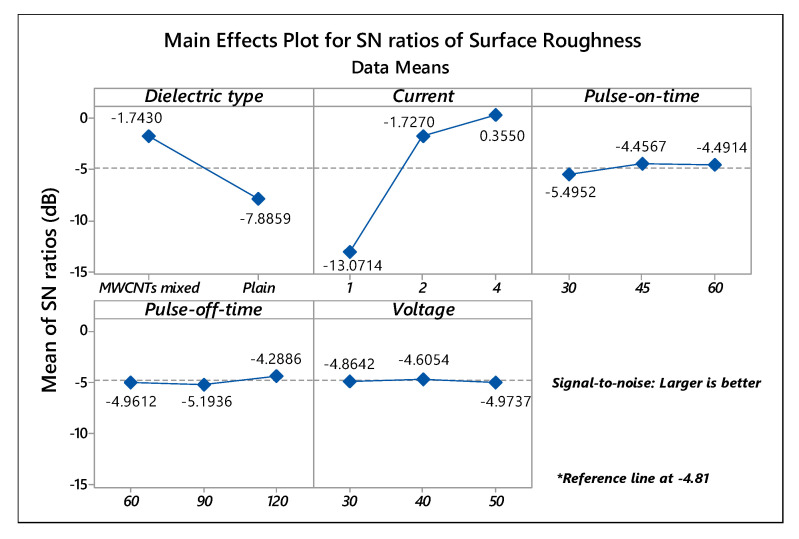
Main effects SN ratios plot of Surface Roughness.

**Figure 4 micromachines-11-00850-f004:**
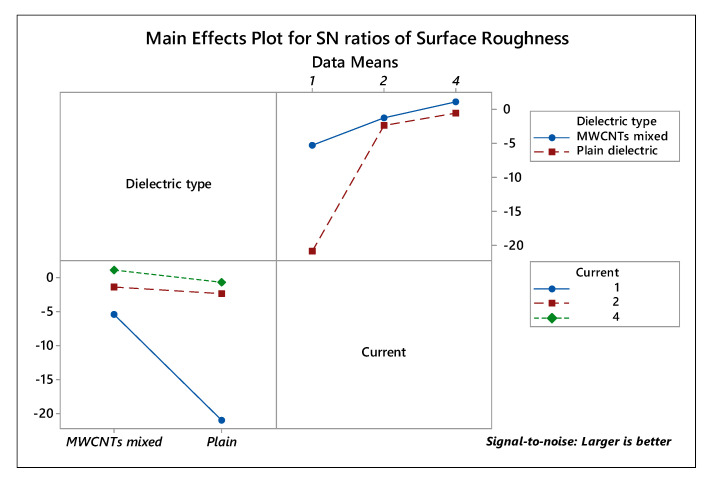
Interaction plot for Surface Roughness.

**Figure 5 micromachines-11-00850-f005:**
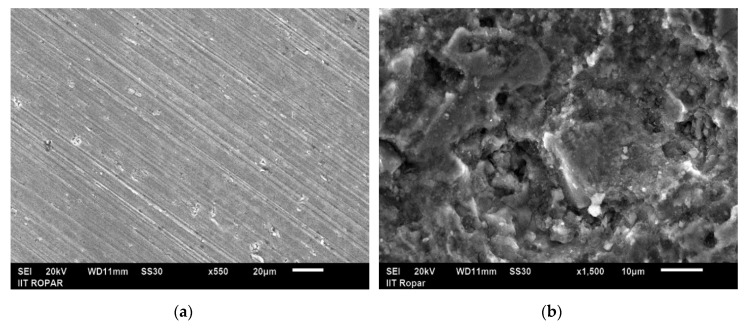
SEM showing; (**a**) substrate or untreated sample, and (**b**) surface treated in MWCNTs mixed dielectric (trial 17) representing micro cracks, porous surface and molten metal droplets.

**Figure 6 micromachines-11-00850-f006:**
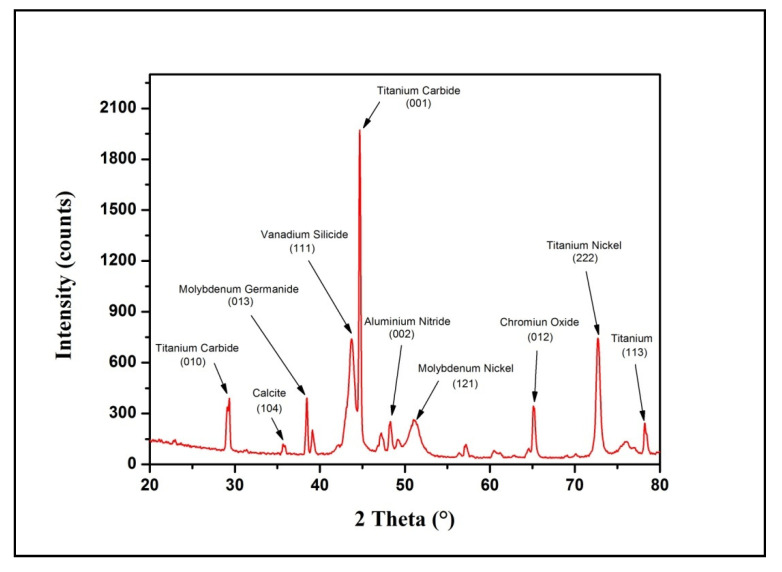
XRD spectra and crystallographic planes of MWCNTs treated sample (trial 17).

**Figure 7 micromachines-11-00850-f007:**
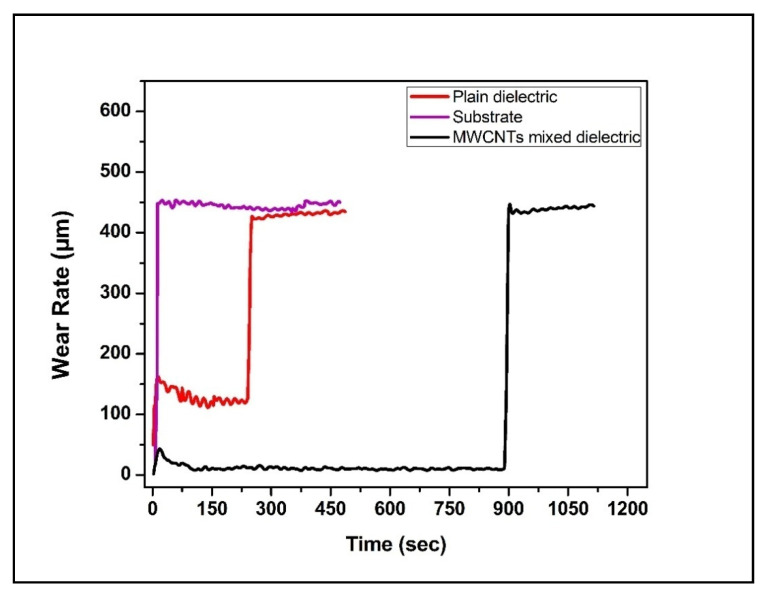
Comparison plot for the wear behavior of specimens.

**Figure 8 micromachines-11-00850-f008:**
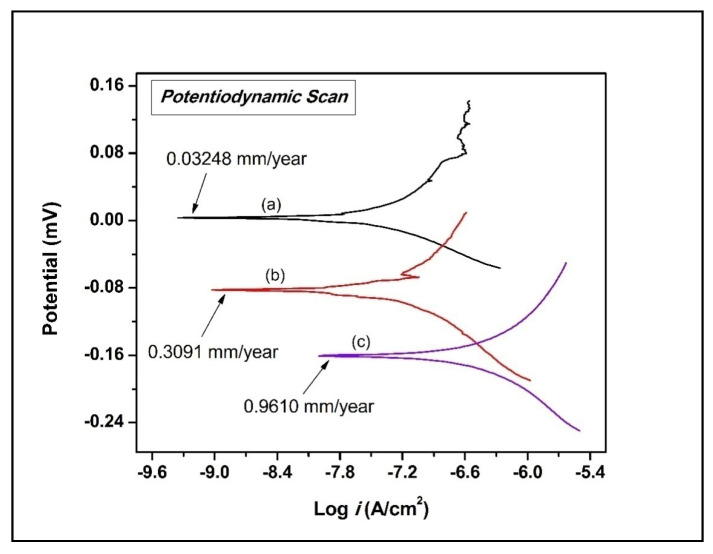
Combined polarization curves of samples (a) treated in MWCNTs, (b) treated in plain dielectric, (c) substrate.

**Table 1 micromachines-11-00850-t001:** Physical properties of workpiece and electrode materials. Source: www.matweb.com.

Property	Ti-6Al-4V	Graphite
Chemical composition	Ti: 89.54%; Al: 6.1%; V: 4.2%; Fe: 0.09%; C: 0.03%; O: 0.03%; N: 0.003%; H: 0.001%	Pure carbon
Size (mm)	70 × 70 × 5	Ø 9.5
Density (g/cm^3^)	4.43	2.26
Melting temperature (°C)	1604–1660	3650
Thermal conductivity (W/m.K)	6.70	24.0
Specific heat (J/Kg °C)	526.3	0.7077
Electrical resistivity (Ω cm)	1.78 × 10^−4^	6.0 × 10^−3^

**Table 2 micromachines-11-00850-t002:** Physical properties of MWCNTs. Source: Technical Data Sheet, provided with powder.

Property	Description
Production Method	Chemical Vapor Deposition
Available form	Black powder
Diameter	Outer Diameter: 10–30 nm
Length	10 microns
Nanotubes purity	>95%
Metal particles	<4%
Amorphous carbon	<1%
Specific surface area	330 m^2^/g
Bulk density	0.04–0.06 g/cm^3^

**Table 3 micromachines-11-00850-t003:** Experimental process parameters with their respective levels.

Parameter	Symbol	Units	Levels		
			Level 1	Level 2	Level 3
Dielectric type	A	–	Plain dielectric	MWCNTs mixed dielectric	–
Current	B	ampere	1	2	4
Pulse-on time	C	µ-seconds	30	45	60
Pulse-off time	D	µ-seconds	60	90	120
Voltage	E	volts	30	40	50

**Table 4 micromachines-11-00850-t004:** Taguchi’s L18 experimental design matrix.

Exp. Trial	Levels of Process Parameters					Actual Values of Process Parameters				
	A	B	C	D	E	A	B	C	D	E
1.	1	1	1	1	1	Plain dielectric	1	30	60	30
2.	1	1	2	2	2	Plain dielectric	1	45	90	40
3.	1	1	3	3	3	Plain dielectric	1	60	120	50
4.	1	2	1	1	2	Plain dielectric	2	30	60	40
5.	1	2	2	2	3	Plain dielectric	2	45	90	50
6.	1	2	3	3	1	Plain dielectric	2	60	120	30
7.	1	3	1	2	1	Plain dielectric	4	30	90	30
8.	1	3	2	3	2	Plain dielectric	4	45	120	40
9.	1	3	3	1	3	Plain dielectric	4	60	60	50
10.	2	1	1	3	3	MWCNTs mixed dielectric	1	30	120	50
11.	2	1	2	1	1	MWCNTs mixed dielectric	1	45	60	30
12.	2	1	3	2	2	MWCNTs mixed dielectric	1	60	90	40
13.	2	2	1	2	3	MWCNTs mixed dielectric	2	30	90	50
14.	2	2	2	3	1	MWCNTs mixed dielectric	2	45	120	30
15.	2	2	3	1	2	MWCNTs mixed dielectric	2	60	60	40
16.	2	3	1	3	2	MWCNTs mixed dielectric	4	30	120	40
17.	2	3	2	1	3	MWCNTs mixed dielectric	4	45	60	50
18.	2	3	3	2	1	MWCNTs mixed dielectric	4	60	90	30

**Table 5 micromachines-11-00850-t005:** Response observations and SN ratios of microhardness (MH) and surface roughness (SR).

Exp. Trial	Output Responses						SN Ratio, dB	
	MH (HV)			SR (µm)			MH	SR
	Rep 1	Rep 2	Rep 3	Rep 1	Rep 2	Rep 3		
1.	897.8	937.0	925.7	0.053	0.061	0.087	59.2731	−24.0190
2.	1066.5	929.3	1025.7	0.138	0.094	0.105	60.0179	−19.3177
3.	1177.4	1098.1	1109.8	0.106	0.101	0.119	61.0373	−19.3390
4.	1091.9	1287.0	1259.3	0.802	0.657	0.883	61.6051	−2.3502
5.	1484.3	1561.2	1586.0	0.661	0.794	0.675	63.7615	−3.0605
6.	1497.0	1434.7	1205.9	0.698	1.021	0.976	62.6768	−1.3158
7.	1419.8	1459.2	1384.3	0.959	1.014	1.018	63.0465	−0.0358
8.	1736.8	1689.1	1656.5	0.855	0.954	0.969	64.5740	−0.7086
9.	1748.3	1761.2	1786.0	0.897	0.892	0.941	64.9347	-0.8266
10.	2140.5	2362.8	2306.1	0.523	0.591	0.615	67.0966	−4.8491
11.	2705.1	2653.1	3067.4	0.534	0.698	0.686	68.9165	−4.0849
12.	2901.8	3242.1	3263.2	0.412	0.469	0.501	69.8885	−6.8189
13.	3147.0	3242.9	3171.3	0.676	0.733	0.709	70.0657	−3.0383
14.	2890.0	2982.9	3329.9	0.885	0.974	0.874	69.6885	−0.8398
15.	3488.2	3792.7	3524.3	1.118	0.980	1.002	71.1123	0.2422
16.	4530.9	4346.8	4306.0	0.991	1.322	1.267	72.8520	1.3207
17.	4394.0	4494.0	4469.6	1.527	0.913	1.280	72.9709	1.2709
18.	3979.4	4283.0	4293.2	1.248	1.226	0.992	72.4180	1.1097

Rep: Repetitions

**Table 6 micromachines-11-00850-t006:** Analysis of variance for signal-to-noise ratios of Microhardness.

Source	DF	Seq SS	Adj MS	F-Value	*p*-Value	% Contribution
Dielectric type	1	304.897	304.897	571.58	0.000 *	83.019
Current	2	50.309	25.155	47.16	0.010 *	13.698
Pulse-on-time	2	5.919	2.959	5.55	0.031 *	1.612
Pulse-off-time	2	0.142	0.071	0.13	0.877	0.038
Voltage	2	1.727	0.863	1.62	0.257	0.472
Residual error	8	4.267	0.533			1.161
Total	17	367.261				100

DF: degrees of freedom; Seq SS: sequential sum of squares; Adj MS: adjusted mean sum of squares, * Significant at 95% confidence level, Rank 1: Dielectric type, Rank 2: Current, Rank 3: Pulse-on-time

**Table 7 micromachines-11-00850-t007:** Analysis of variance for signal-to-noise ratios of Surface Roughness.

Source	DF	Seq SS	Adj MS	F-Value	*p*-Value	% Contribution
Dielectric type	1	169.81	169.806	53.88	0.000 *	16.551
Current	2	626.60	313.299	99.41	0.000 *	61.076
Pulse-on-time	2	4.18	2.088	0.66	0.550	0.407
Pulse-off-time	2	2.65	1.325	0.42	0.675	0.258
Voltage	2	0.43	0.215	0.07	0.935	0.041
Dielectric type × current	2	203.38	101.689	32.27	0.001 *	19.824
Residual error	6	18.91	3.151			1.843
Total	17	1025.95				100

DF: degrees of freedom; Seq SS: sequential sum of squares; Adj MS: adjusted mean sum of squares, * Significant at 95% confidence level, Rank 1: Current, Rank 2: Dielectric type, Rank 3: Dielectric type × current

**Table 8 micromachines-11-00850-t008:** Polarization corrosion data of specimens in Ringer solution at 37 °C.

Sr. No.	Sample	Ecorr (mV)	*i*_corr_ (µA/cm^2^)	βa (mV/dec)	βc (mV/dec)	Corrosion Rate (mm/y)	Protection Efficiency (Pe)
1.	MWCNTs mixed dielectric	3.50860	1.83	93.1230	187.420	0.03248	96.63%
2.	Plain dielectric	−82.4920	17.48	211.820	126.560	0.3091	67.83%
3.	Substrate	−160.640	54.35	234.610	292.470	0.9610	–

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
