# Peer review of "Enhancing Corrosion and Wear Resistance of Ti6Al4V Alloy Using CNTs Mixed Electro-Discharge Process"

_micromachines, 2020, doi:10.3390/mi11090850_

Round 1
Reviewer 1 Report
The authors present an interesting study about the use of MWCNTs on Ti6Al4V alloys processed by electro-discharge treatment. This leads to change of corrosion and wear resistance. In general, the study is well prepared, however, there is a question that remain to be answered before publication can be recommended.
Are there any risks regarding MWCNTs on a surface in contact with human tissue?
Additionally, authors should make the following corrections:
Fig. 1.: The meanings 'a' and 'b' are missing in the figure
Line 160: '.... with better surface resposes' - what the author had in mind?
Line 210: Add (S) and (N) in parentheses as first used.
Table 5: 'MH, SR' - use abbreviations earlier in the text or in the next column.
Figure 2, Figure 3: 'SN' or S/N ratio on the x and y axes?
Line 280: 'It was evident from the micrographs that higher spark energy coupled with carbon particles produces micro-macropores, small peaks, and valleys on the EDT surface.' As compared to what? Substrate, plain dielectric? In order to make such a statement, one should present the microstructure 'surface treated in plain dielectric' or/and 'substrate'.
Figure 5: Subfigure 'a' and 'b are very similar, no obvious differences, can show the surface at a different magnification if it brings something new? The porous structure is already visible in subfigure 'a'.
Line 286: What is the difference between 'surface in MWCNTs mixed dielectric' and 'surface in MWCNTs-EDT'? Is it the same sample?
Figures 7 and 9: All curves should be legible in black and white. The same color (line style) should be applied to the same sample.
Figure 9: Double bracket in the x axis description.
Line 379: '...which facilitates the proper bone-implant adhesion and promotes cell proliferation.' There are no studies on cell adhesion and proliferation in this article. The authors can only assume that the obtained structure will affect the above-mentioned properties.
In my opinion the [33] self-citations is inappropriate. Acceptable of roughness value for the bio-implants for proper cell proliferation, bone-implant fixation comes from other publications cited as [36, 37] in the discussed citation [33].
I suggest adding the full name of the alloy (Ti6Al4V) in the manuscript title.
Author Response
Observation 1
Are there any risks regarding MWCNTs on a surface in contact with human tissue?
MWCNTs are multi-walled graphene sheets rolled to form carbon nanotubes. Numerous papers have been already published that graphene or carbon nanotubes depicts a biofriendly behavior with the human tissues. Some references are added in the revised manuscript to support the comment.
Description: -
- George, G., Sisupal, S.B., Tomy, T. et al.Facile, environmentally benign and scalable approach to produce pristine few layers graphene suitable for preparing biocompatible polymer nanocomposites. Scientific Reports 8, 11228 (2018). https://doi.org/10.1038/s41598-018-28560-1
- Kumar, S., Parekh, S.H. Linking graphene-based material physicochemical properties with molecular adsorption, structure and cell fate. Communications Chemistry3, 8 (2020). https://doi.org/10.1038/s42004-019-0254-9
Observation 2
Fig. 1.: The meanings 'a' and 'b' are missing in the figure.
The authors regret typos. The concern figure is corrected and updated accordingly.
Observation 3
Line 160: '.... with better surface responses' - what the author had in mind?
It was reported in the literature that higher surface roughness participated in the cell anchoring providing better bone-implant adhesion interface due to the presence of micro-crack and pores [1-3]. Therefore, for further testing, the samples with higher surface roughness were taken out from the workpiece plate using wire-EDM.
The changes are highlighted in the revised manuscript; also references are added to support the comment.
Description: -
- Buser, D., Schenk, R. K., Steinemann, S., Fiorellini, J. P., Fox, C. H., & Stich, H. (1991). Influence of surface characteristics on bone integration of titanium implants. A histomorphometric study in miniature pigs. Journal of Biomedical Materials Research, 25(7), 889–902. doi:10.1002/jbm.820250708
- Larsson, C., Thomsen, P., Aronsson, B.-O., Rodahl, M., Lausmaa, J., Kasemo, B., & Ericson, L. E. (1996). Bone response to surface-modified titanium implants: studies on the early tissue response to machined and electropolished implants with different oxide thicknesses. Biomaterials, 17(6), 605–616. doi:10.1016/0142-9612(96)88711-4
- Mour, M.; Das, D.; Winkler, T.; Hoenig, E.; Mielke, G.; Morlock, M.M.; Schilling, A.F. Advances in Porous Biomaterials for Dental and Orthopaedic Applications. Materials 2010, 3, 2947-2974. doi: https://doi.org/10.3390/ma3052947
Observation 4
Line 210: Add (S) and (N) in parentheses as first used.
The manuscript is checked, and updated for all the ‘SN ratio’ words throughout the manuscript. The needful corrections have been done and changed accordingly.
Observation 5
Table 5: 'MH, SR' - use abbreviations earlier in the text or in the next column.
The heading of the table 5 is updated i.e. microhardness (MH) and surface roughness (SR).
Observation 6
Figure 2, Figure 3: 'SN' or S/N ratio on the x and y axes?
It is SN ratio i.e. signal-to-noise ratio. The error is corrected and highlighted in the revised submission.
Observation 7
Line 280: 'It was evident from the micrographs that higher spark energy coupled with carbon particles produces micro-macropores, small peaks, and valleys on the EDT surface.' As compared to what? Substrate, plain dielectric? In order to make such a statement, one should present the microstructure 'surface treated in plain dielectric' or/and 'substrate'.
The SEM micrographs (figure 5) are updated in the revised manuscript. A SEM image is updated with the untreated substrate SEM image, and the electro-discharge treated sample for better comparison. The changes are highlighted (in red color) in the manuscript as follow.
Observation 8
Figure 5: Subfigure 'a' and 'b’ are very similar, no obvious differences, can show the surface at a different magnification if it brings something new? The porous structure is already visible in subfigure 'a'.
The figure 5b is replaced with the untreated substrate SEM, and updated SEM micrographs are, (a) substrate and (b) MWCNTs-EDT sample.
Observation 9
Line 286: What is the difference between 'surface in MWCNTs mixed dielectric' and 'surface in MWCNTs-EDT'? Is it the same sample?
Both the words represent the same thing i.e. the machining in MWCNTS mixed dielectric as flushing medium. However, the caption of figure 5 is updated to eliminate the confusion.
Description: -
Figure 5. SEM showing (a) substrate or untreated sample, and (b) surface treated in MWCNTs mixed dielectric (trial 17) representing micro cracks, porous surface and molten metal droplets.
Observation 10
Figures 7 and 9: All curves should be legible in black and white. The same color (line style) should be applied to the same sample.
Both the figures are updated and same colours applied to the same type of sample i.e. treated in MWCNTs (black color), treated in plain dielectric (red color) and untreated substrate (purple color).
Observation 11
Figure 9: Double bracket in the x axis description.
The concern figure is revised and modified accordingly.
Observation 12
Line 379: '...which facilitates the proper bone-implant adhesion and promotes cell proliferation.' There are no studies on cell adhesion and proliferation in this article. The authors can only assume that the obtained structure will affect the above-mentioned properties.
According to the similar studies by the other researchers, the surface depicting porous structure, intermetallic bioactive compounds, enhanced wear resistance and corrosion resistance is prominent for cell growth and bone regeneration. The results of present work are in-accordance with these studies and thus a remark is concluded that the MWCNTs-EDT surface facilities the proper bone-implant adhesion and promotes cell proliferation.
However, to avoid the confusion and to justify the comment, the sentence is revised and word ‘may’ is added in the revised manuscript. Also, the references for each observation is already in the relevant sections.
Description: -
“SEM analysis disclosed that MWCNTs-EDT surface with the presence of small craters, and evenly distributed micro-pores on the surface which may facilitates the proper bone-implant adhesion and promotes cell proliferation”.
Observation 13
In my opinion the [33] self-citation is inappropriate. Acceptable of roughness value for the bio-implants for proper cell proliferation, bone-implant fixation comes from other publications cited as [36, 37] in the discussed citation [33].
New references have been added to support the comment.
Description: -
- Melentiev, R., Kang, C., Shen, G., & Fang, F. (2019). Study on surface roughness generated by micro-blasting on Co-Cr-Mo bio-implant. Wear, 428-429, 111–126. doi: 1016/j.wear.2019.03.005
- Rao, S., Hashemiastaneh, S., Villanueva, J., Silva, F., Takoudis, C., Bijukumar, D., … Mathew, M. T. (2019). In vitro osseointegration analysis of bio-functionalized titanium samples in a protein-rich medium. Journal of the Mechanical Behavior of Biomedical Materials. doi: 1016/j.jmbbm.2019.03.019
- De Bruyn, H., Christiaens, V., Doornewaard, R., Jacobsson, M., Cosyn, J., Jacquet, W., & Vervaeke, S. (2016). Implant surface roughness and patient factors on long-term peri-implant bone loss. Periodontology 2000, 73(1), 218–227. doi: 1111/prd.12177
Observation 14
I suggest adding the full name of the alloy (Ti6Al4V) in the manuscript title.
The suggestion of the reviewer is appreciable, and the title is updated accordingly.
Authors sincerely thank the reviewer(s) for their valuable comments and suggestions that helped to
improve the quality of the paper.
Thanks & Regards
Reviewer 2 Report
- Please explain the method to disperse MWCNTs and provide some characterisation regarding the resulting structure.
- To compare the mechanical and electrochemical performance, the cross-sectional microstructure should be provided.
- In terms of such surface modification method, binding force between the surface layer and the substrate might be an issue, especially for the biological application. Could you please explain your relevant consideration?
- To confirm the phases that may be formed during the process, the element analysis must be further performed. In addition, could you please explain the possible reason related to the formation of intermetallic compounds?
- For biological application, there are some elements which are harmful to human body. Could you please explain your relevant consideration?
- In general, the interconnected pores with the dimension mainly in the range of 300~600 microns are meaningful regarding biological application. Could you please provide more information and characterisation?
Author Response
Observation 1
Please explain the method to disperse MWCNTs and provide some characterisation regarding the resulting structure.
In the reported work, hydrocarbon carbon was used as dielectric fluid and MWCNTs was dispersed using the stirrer mounted on the fabricated tank. As our objective was to properly circulate MWCNTs mixed dielectric in the machining area, a pump was also there for proper supply of dielectric. However, we have not studied the change in the characterization of the resulting structure as it doesn’t affect the machining process.
We have already described the powder disperse phenomenon in the experimental work section, and highlighted in red. Also figure 1 is clearly describing the process.
Description: -
A circulation pump and stirrer were introduced for efficient flushing, supply of nano-powder in the spark gap during machining, and homogeneous mixing of dielectric medium.
Figure 1. EDT line-diagram in (a), and indigenous setup for MWCNTs-EDT in (b).
Observation 2
To compare the mechanical and electrochemical performance, the cross-sectional microstructure should be provided.
The cross-sectional microstructure evaluation related to mechanical and electrochemical performance comparison is not performed in the present study, but it may be extended in the future research work.
Observation 3
In terms of such surface modification method, binding force between the surface layer and the substrate might be an issue, especially for the biological application. Could you please explain your relevant consideration?
In the present work, scratch test is not performed to examine the adhesion. It will be done in the extension of this study along with the investigation of recast thickness and residual stresses analysis. Wear test is reported in this work to relate the coated surface with substrate material for wear resistance examination. However, in the past studies it has been confirmed by the authors that the surface formed by the electro-discharge treatment are harder and sufficient binding force was developed for sustainable performance of implants.
Reference: -
Devgan, S., Sidhu, S. S. (2019). Enhancing Tribological Performance of β-Titanium Alloy Using Electrical Discharge Process. Surface Innovations, 8(1-2), 115–126. doi:10.1680/jsuin.19.00031
Observation 4
To confirm the phases that may be formed during the process, the element analysis must be further performed. In addition, could you please explain the possible reason related to the formation of intermetallic compounds?
It is good idea to add elemental analysis (EDS) with the XRD spectra, but due to the limitation to access the lab, we are not able to perform EDS. However, the reason behind the formation of intermetallic compounds is the high spark energy generated during the process. The elevated temperature in the machining region results in the reaction between workpiece elements, electrode material (graphite), the carbon nanotubes, and the hydrocarbon oil. The breakdown of elements and their reaction with other elements form the intermetallic compounds, carbides and silicide.
Observation 5
For biological application, there are some elements which are harmful to human body. Could you please explain your relevant consideration?
The presence of aluminium and vanadium in Ti-6Al-4V is harmful in long-term implantation of these bioimplants. But when the surface is treated at high spark energy, the formation of carbides and silicides restricts the decomposition of aluminium and vanadium from the surface of implants. It acts as protective layer and safeguard from the harmful ions release due to the decomposition of aluminium and vanadium from bioimplant.
However, the rate of degradation of implants after EDT may be included in future scope
Observation 6
In general, the interconnected pores with the dimension mainly in the range of 300~600 microns are meaningful regarding biological application. Could you please provide more information and characterisation?
In our study the pores distribution and size are produced due to the spark generated between workpiece and electrode. However, in order to study the pore size or interconnection of pores on this material, we have not done the cross-sectional examination. It can be performed in various in-vivo and in-vitro tests related to the cell viability.
Authors sincerely thank the reviewer(s) for their valuable comments and suggestions that helped to
improve the quality of the paper.
Thanks & Regards

Round 2
Reviewer 2 Report
The response is acceptable.